# Predicted Membrane-Associated Domains in Proteins Encoded by Novel Monopartite Plant RNA Viruses Related to Members of the Family *Benyviridae*

**DOI:** 10.3390/ijms241512161

**Published:** 2023-07-29

**Authors:** Sergey Y. Morozov, Alexander A. Lezzhov, Andrey G. Solovyev

**Affiliations:** 1A. N. Belozersky Institute of Physico-Chemical Biology, Moscow State University, 119992 Moscow, Russia; lezzhov-genetic@mail.ru (A.A.L.); solovyev@belozersky.msu.ru (A.G.S.); 2Department of Virology, Biological Faculty, Moscow State University, 119234 Moscow, Russia

**Keywords:** RNA virus, benyviruses, RNA polymerase, zinc-finger motifs, membrane-embedded protein segments, virus-movement proteins, movement system evolution

## Abstract

As a continuation of our previous work, in this paper, we examine in greater detail the genome organization and some protein properties of the members of a potential group named Reclovirids and belonging to *Benyviridae*-related viruses. It can be proposed that the single-component Reclovirid genomes encode previously undiscovered transport genes. Indeed, analysis of the coding potential of these novel viral genomes reveals one or more cistrons ranging in size from 40 to 80 to about 600 codons, located in the 3′-terminal region of the genomic RNA, encoding proteins with predicted hydrophobic segments that are structurally diverse among Reclovirids and have no analogues in other plant RNA viruses. Additionally, in many cases, the possible methyltransferase domain of Reclovirid replicases is preceded by membrane-embedded protein segments that are not present in annotated members of the *Benyviridae* family. These observations suggest a general association of most Reclovirid proteins with cell membranes.

## 1. Introduction

Some recent papers have highlighted a significant diversity of monopartite plant, fungal, and insect viruses encoding replicases related to those encoded by the multipartite RNA genomes of members of the genus *Benyvirus* [1,2,3,4,5,6].

However, RNA-dependent RNA polymerase (RdRp) sequences from viruses of the genus *Benyvirus* are most closely related to the proteins encoded by VLRAs of land plants only, and form a distinct branch of the corresponding phylogenetic tree (see Figure 2 in [2] and Figure 3 in [6]). This cluster of the phylogenetic tree includes two sub-branches, namely those representing viruses with multipartite and monopartite genomes. In addition to members of the genus *Benyvirus*, the former branch includes several members of the *Benyviridae* family, particularly, *Wheat stripe mosaic virus* and *Fern benyvirus*. It also contains a recently described group of “Tetra-cistron movement block (TCMB)-containing viruses” (Tecimovirids) coding for the TCMB movement gene module instead of the triple gene block (TGB) of movement genes found in members of the family *Benyviridae*. [2,7] (Figure 1). The latter branch contains exclusively monopartite viruses and, in particular, includes *Goji berry chlorosis virus* (GBCV), which encodes six polypeptides and has no close relatives with similar genome organization [8]. Four additional benyvirus-like members of this brunch are predicted to encode a binary movement block (BMB), which is also found in multipartite viruses of the family *Kitaviridae* [7,9,10,11] (Figure 1).

In the course of recent studies on the phylogeny of benyvirus-like RNA polymerases, we found that phylogenetically close monopartite viruses include a new, previously undescribed group of viruses related to members of the family *Benyviridae*. These viruses contain polyadenylated, single-component RNA genomes with a maximum size of up to 10,000 nucleotide residues, include several species annotated in the NCBI database, and have been named Reclovirids after *Red clover virus* 1 [2]. In contrast to the vast majority of annotated plant viruses in the family *Benyviridae*, the viral genomes of Reclovirids do not encode any previously characterized transport gene blocks. Our initial assumption that these viral genomes belong to two- or multi-component viruses has not been confirmed [2]. Thus, it would be proposed that the single-component Reclovirid genomes encode previously undiscovered transport genes. The coding potential of these novel viral genomes has revealed one or more cistrons ranging in size from 40 to 80 to about 600 codons, located in the 3′-terminal region of the genomic RNA. The encoding proteins with predicted hydrophobic segments have no analogues in other plant RNA viruses [2]. These data allowed us to hypothesize that flowering plants can be infected with the novel viruses (Reclovirids), which are related to the members of family *Benyviridae* and belong to a new taxon in the order *Hepelivirales* (probably a subfamily, or even a family). In this paper, we comparatively analyzed the gene organization of Relovirid genomes and the structural properties of the encoded non-replicative “orphan” hydrophobic polypeptides as well as replicative polypeptides.

## 2. Results

The NCBI non-redundant nucleotide sequence library currently contains the only four annotated Reclovirids. These viruses include *Red clover virus* 1, *Dactylorhiza hatagirea beny-like virus*, *Carrot associated RNA virus* 1, and *Arceuthobium sichuanense virus* 3 [2]. Public databases of plant transcriptomes proved to be a good source for identifying previously unknown viruses. Therefore, we searched the NCBI transcriptome shotgun assembly (TSA), short read assembly (SRA) and 1KP libraries for Reclovirus-like transcripts in order to identify novel viral sequences. We have discovered a total of 27 viruses (Figure 1) infecting 78 species from 12 families of monocotyledonous and dicotyledonous flowering plants (Table 1).

### 2.1. Genome Organization of Reclovirids

Most Reclovirid genomes contain two ORFs, namely, the 5′-terminal long replicase ORF (approximate size of 6300–7000 nucleotides in the nearly full-length genomic RNAs) and the shorter downstream ORF with sizes varying from 330 to 2000 nts [2] (Appendix A). In some cases, Reclovirid genomes encode two proteins in addition to replicase, and their ORFs often overlap. An exception is *Scutellaria montana* VLRA, which encodes four small proteins in addition to replicase (Figure 2) (Appendix A). The gene organization of the 3′-terminal regions in Reclovirid RNAs is rather variable. For example, two highly similar VLRAs found in *Gymnadenia rhellicani* (family *Orchidaceae*) (Table 1), which are closely related in the RdRp phylogenetic tree (Figure 1), show different numbers of the 3′-terminal ORFs (Figure 2) (Appendix A).

### 2.2. Protein Domains in the Replicases of Reclovirids

We have previously reported that the RNA polymerases of *Red clover RNA virus* 1 and other Reclovirids have two characteristic domains, namely, a viral helicase 1 domain (HEL, pfam01443) and an RdRp core motif (pfam00978). In addition, analysis of the NCBI Conserved Domain Database (CDD) has shown no viral methyltransferase domain (MTR, pfam01660) in the replicases of Reclovirids as well as members of the genus *Benyvirus* [2]. Nevertheless, the current comprehensive CDD analysis of benyvirus-like replicases distantly related to those of Reclovirids (see Figure 2 in [2]) has shown that the insect *Hubei beny-like virus* 1 (HBLV1) encodes an MTR domain (pfam01660, e-value 6e−16) in the N-terminal part of the replicase protein. The BLASTP search revealed that Reclovirid full-length replicases contain moderately comparable domains of 340–350 amino acids (Table 2 and Figure 2), which yield negative CDD results. Pairwise comparisons revealed that all Reclovirid MTR sequences contain a number of gaps when compared to the HBLV1 protein domain (Table 2). This fact may lead to negative results in CDD searches for Reclovirid replicases. Importantly, a recent general comparative analysis of *Riboviria*-encoded methyltransferases [12] clearly indicates that the methyltransferase domains are characteristic for replicases of *Benyviridae* and particularly the genus *Benyvirus*.

Interestingly, in some cases, the possible methyltransferase domain of Reclovirid replicases is preceded by membrane-embedded protein segments that are not present in annotated members of the family *Benyviridae* (Appendix A). To date, the presence and function of membrane-spanning segments in viral RNA replicases has only been well-characterized for coronaviruses [13,14].

### 2.3. Protein Domains and Motifs in Non-Replicative Proteins of Reclovirids

It has been shown that most non-replicative proteins of Reclovirids possess predicted membrane-embedded segments [2] (Appendix A). The length of these proteins varies from 40 to nearly 670 amino acids (Appendix A). All of these proteins represent “orphan” viral proteins with membrane-spanning domains [9]. Previously, pairwise sequence comparisons have revealed a group of hydrophobic non-replicative proteins with apparent overall sequence conservation among Reclovirids infecting the family *Orchidaceae* (see Figure 9 in [2]). All these proteins contain a characteristic CX(3)CX(10)CX(3)C motif (putative zinc-finger domain) in the N-terminal region (Figure 3) and hydrophobic segments in the C-terminal half (Appendix A). Currently, we have found putative zinc-finger domains with slightly different motifs in non-replicative proteins of some other Reclovirids (Figure 3A).

Quite interestingly, CDD analysis of non-replicative proteins of Reclovirids revealed that the largest of these proteins (668 aa in length), the *Astragalus canadensis* VLRA ORF2 protein, contains a domain of the Mpp10 protein family (COG5384) (positions 205–465, e-value 2.84e−03). This family includes proteins related to Mpp10, which is part of the U3 small nucleolar ribonucleoprotein in yeast [15]. Outside the Mpp10-like domain, this protein contains two putative membrane-embedded segments (Appendix A) and a region of distant similarity (identity 24%, e-value 2.84e−03) to the non-replicative protein of two other Reclovirids (namely, 2063722-*Leontopodium_alpinum* VLRA and *Vicia faba* VLRA) (Appendix A). This region contains a putative zinc-finger domain with the signature CX(3)CX(7)CX(3)C (Figure 3B).

Our analysis revealed that a number Reclovirids encode non-replicative proteins with other types of putative zinc-finger motifs. In particular, the ORF2 protein of *Atriplex prostrata* VLRA (Appendix A) contains the hexa-cysteine motif C(X6)C(X)C(XX)CXC(X12)C, which resembles unconventional hexa-cysteine motifs like those found in proteins of *Hepatitis virus E* and viruses of the genus *Pestivirus* [16,17], and in the small protein encoded by the ORF preceding TCMB in *Colobanthus quitensis* VLRA [7].

## 3. Discussion

The discovery of multiple plant-specific, capsidless Reclovirid VLRAs by high-throughput sequencing raises a number of questions about the evolution of the *Benyviridae*-like viruses and, in particular, the directions of evolution of plant virus-movement protein systems [2]. We have proposed that Reclovirids are likely to use novel, as yet undescribed, movement protein systems. The closely related plant *Benyviridae*-like viruses use movement gene blocks (BMB, TGB, or TCMB) that encode one, two or three small proteins with hydrophobic membrane-spanning segments as well as RNA helicase. On the other hand, the Reclovirid cell-to-cell movement may be carried out by a wide variety of the hydrophobic “orphan” proteins (usually a single protein per virus) [2].

What characteristics of these proteins make them suitable candidates for viral RNA transmission from cell to cell? What are the specific features that make these proteins suitable candidates for performing cell-to-cell trafficking of virus RNA? Considering single MP-based transport systems exemplified by that of Tobacco mosaic virus, several features can be distinguished, namely, the ability to bind RNA, interact with ER/actin/microtubules, modify plasmodesmata (PD), move from cell to cell, and direct virus replication complexes to PD [18,19]. It should be noted that in case of Reclovirids, the latter function can be performed directly by the replication protein due to the presence of the N-terminal membrane-spanning segments, which can potentially target the ER at the PD entrance. Computer prediction methods have indicated that non-replicative proteins of Reclovirids can perform at least two of the other MP functions. First, most Reclovirid “orphan” proteins contain membrane-spanning segments at their N-termini and internal trans-membrane regions that may direct these proteins to the ER membranes. Second, these proteins possess putative zinc-finger motifs, which in many cases are known to participate in nucleic acid binding (including ssRNA binding) [20,21,22]. Interestingly, the *Rice yellow mottle virus* protein with movement and silencing suppression functions has been shown to contain two essential zinc-finger motifs [23,24,25]. One of these motifs belongs to the C4 type, as those found in most Reclovirid “orphan” proteins, and performs an unknown function in cell-to-cell movement [23]. Thus, it can be proposed that the hydrophobic Reclovirid proteins with the zinc-finger motif may be responsible for genomic RNA binding and the interaction with the ER tubule, as well as increasing the PD permeability. Obviously, these hypotheses require more experimental validation. In evolutionary terms, this type of putative movement protein may represent an alternative for adaptation of beny-like viruses to colonize multicellular land plants.

## 4. Materials and Methods

Reclovirid-related nucleotide and protein sequences were collected from the NCBI plant transcriptome database. Sequence comparisons were carried out using the BLAST algorithm (TBLASTn and BLASTp) at the National Center for Biotechnology Information (NCBI). The nucleotide raw sequence reads from each analyzed SRA experiment linked to the virus nucleotide sequence and TSA projects returning Reclovirid-like hits were downloaded and subjected to bulk local BLASTX searches (e-value ≤ 1e−105) against a Refseq virus protein database available at ftp://ftp.ncbi.nlm.nih.gov/refseq/release/viral/viral.1.protein.faa.gz (accessed on 20 May 2023). The resulting viral sequence hits of each SRA read were then visually explored. Tentative virus contigs were extended by iterative mapping of each SRA library’s raw reads. This strategy employed re-iterative BLAST to extract a subset of reads related to the query contig, and these retrieved reads were used to extend the contig, and then, the process was repeated iteratively using as the query the extended sequence.

ORFs were identified using the ORF Finder programs (https://www.ncbi.nlm.nih.gov/orffinder/, accessed on 20 May 2023). Conserved motif searches were conducted in CDD (http://www.ncbi.nlm.nih.gov/Structure/cdd/wrpsb.cgi, accessed on 20 May 2023) databases. Membrane-embedded hydrophobic protein regions were predicted using the software TOPCONS [26] (https://topcons.cbr.su.se/pred/, accessed on 20 May 2023). Multiple sequence alignment was performed using NCBI software (https://www.ncbi.nlm.nih.gov/tools/cobalt/, accessed on 20 May 2023). Phylogenetic analysis was performed with “Phylogeny.fr” (a free, simple-to-use web service dedicated to the reconstruction and analysis of phylogenetic relationships between molecular sequences) by constructing maximum likelihood phylogenetic trees (http://www.phylogeny.fr/simple_phylogeny.cgi accessed on 20 May 2023). Bootstrap percentages received from 1000 replications were used. Genome sequences of different benyvirus viruses were downloaded from the GenBank database.

## 5. Conclusions

In conclusion, it should be noted that a number of the recently annotated plant tymovirus-like genomes may have small “orphan” hydrophobic protein ORFs and lack the well characterized CPs and MP systems. In particular, these viruses include *Broom forkmoss associated tymo-like virus*, *Yellow horn associated tymo-like virus*, *Badge moss associated tymo-like virus*, *Polish wheat virus* 1, *Kava virus* 1, *Agave tequiliana deltaflexivirus* 1, and *Sesame deltaflexivirus* 1 [6,27]. Thus, it appears that Reclovirids represent members of a class of plant viruses with missing or as yet unknown cell-to-cell movement systems. Indeed, it is known that persistent plant viruses lack cell-to-cell movement systems and do not cause visible symptoms; accordingly, they are transmitted only vertically via gametes. Persistent plant viruses represent a few virus families such as *Endornaviridae* [28]. Among non-persistent plant viruses, several viruses named umbravirus-like associated RNAs (ulaRNAs) that lack both CP and MP genes have recently been discovered [29,30]. It is proposed that ulaRNAs could be transmitted by arthropods and spread within the plant body with the help of other co-infecting viruses from the family *Tombusviridae*. After the initial vector transmission, the helper virus could be lost during progress of infection, for example, because of high temperatures [29].

## Figures and Tables

**Figure 1 ijms-24-12161-f001:**
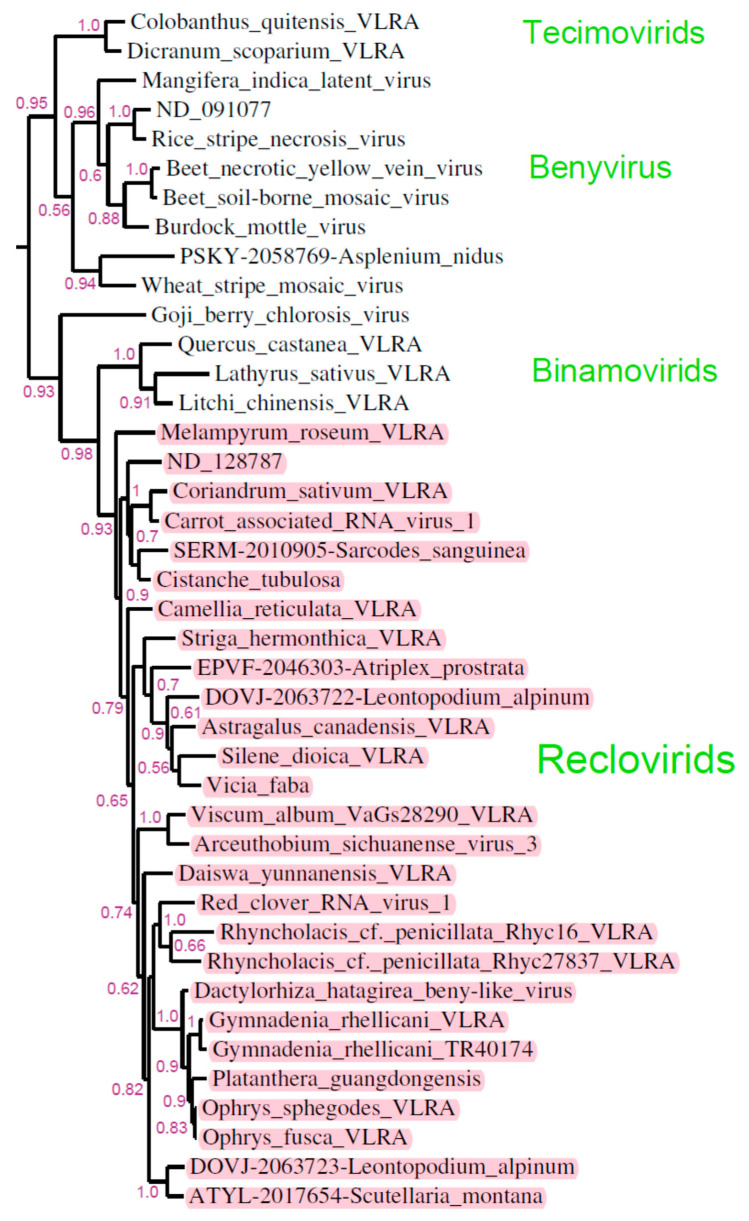
Updated phylogenetic subtree of the beny-like RdRp domains showing a cluster of plant-specific viruses which are significantly related to members of genus Benyvirus. Whole tree was generated using the maximum likelihood method. The bootstrap values obtained with 1000 replicates are indicated on the branches, and branch lengths correspond to the branch line’s genetic distances. Reclovirids are shown in pink.

**Figure 2 ijms-24-12161-f002:**
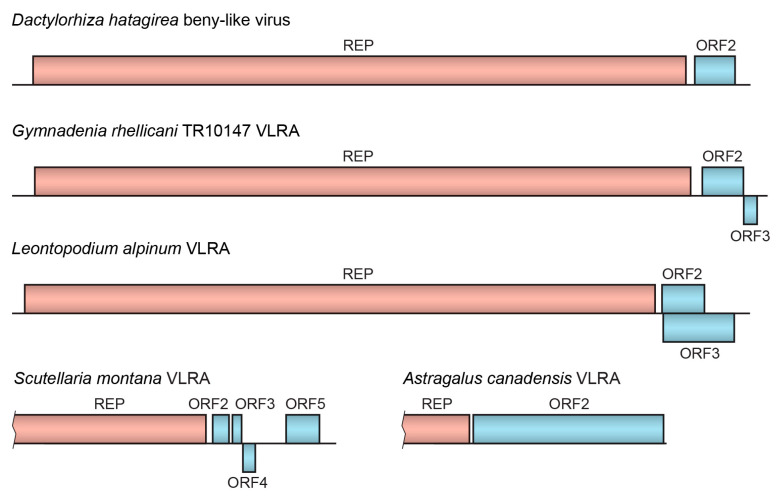
Gene organization of the selected Reclovirids. The replicase protein domains from partial and full-length VLRAs (virus-like RNA assemblies) are shown (see text for details). Proteins with putative membrane-bound segments are shown in blue. ORF means open reading frame, and REP means replicase gene.

**Figure 3 ijms-24-12161-f003:**
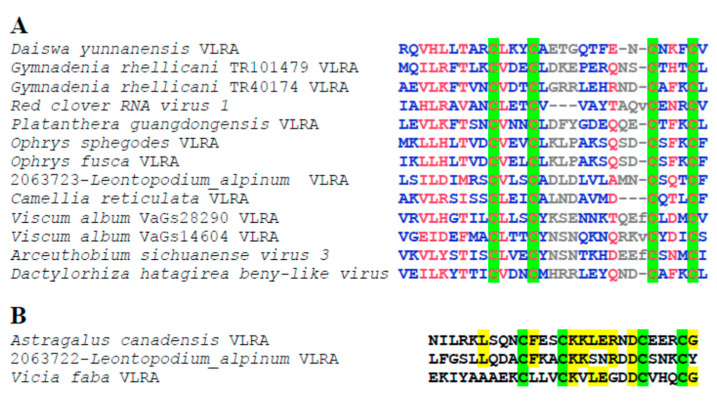
Nucleotide sequence alignments of the putative zinc-finger motifs of ORF2 proteins encoded in the 3′-terminal regions of Reclovirid genomes. (**A**) Multiple sequence alignment using NCBI COBALT software. (**B**) Pairwise comparison by visual inspection. Conserved cysteines in the protein sequences are in green.

**Table 1 ijms-24-12161-t001:** A list of the host plants for viruses representing a group of Reclovirids.

Plant Order/Family	Virus Host Plant	Accession Number
*Asparagales*; *Orchidaceae*	*Dactylorhiza hatagirea*	BK013327
	*Gymnadenia rhellicani*	GHXH01324014
	*Gymnadenia rhellicani*	GHXH01128483
	*Platanthera guangdongensis*	SRX14997078
	*Ophrys sphegodes*	GHXJ01414654
	*Ophrys fusca*	GHXI01129489
*Liliales*; *Melanthiaceae*	*Daiswa yunnanensis*	GFOY01013898
*Ericales*; *Theaceae*	*Camellia reticulate*	GEER01003429
*Ericales*; *Ericaceae*	*Sarcodes sanguinea*	SERM-2010905 *
*Asterales*; *Asteraceae*	*Leontopodium alpinum*	DOVJ-2063723 *
	*Leontopodium alpinum*	DOVJ-2063722 *
*Lamiales*; *Lamiaceae*	*Scutellaria montana*	ATYL-2017654 *
*Lamiales*; *Orobanchaceae*	*Cistanche tubulosa*	GJRS01079843
	*Melampyrum roseum*	IADV01103213
	*Striga hermonthica*	ICPL01009187
*Apiales*; *Apiaceae*	*Daucus carota*	OM419188; SRX13122999
	*Coriandrum sativum*	GGPN01001998
*Caryophyllales*; *Chenopodiaceae*	*Atriplex prostrata*	AAXJ-2011446 *
	*Silene dioica*	GFCG01071918
*Santalales*; *Viscaceae*	*Arceuthobium sichuanense*	BK059270
	*Viscum album*	GJLG01028288
	*Viscum album*	GJLG01014603; SRX12291946
*Malpighiales*; *Podostemaceae*	*Rhyncholacis* cf. penicillata	ICSC01000014
	*Rhyncholacis* cf. penicillata	ICSC01056734
*Fabales*; *Fabaceae*	*Vicia faba*	GISP01006645; SRX10153333
	*Astragalus canadensis*	GGNK01006218
	*Trifolium pretense*	MG596242

* The accession number refers to the 1KP database.

**Table 2 ijms-24-12161-t002:** A list of the selected Reclovirid replicases having homology to HBLV1 MTR domain.

VLRA, Accession Number	Percentage of Identity/Gaps	Domain Position *, E-Value
*Dactylorhiza hatagirea virus*, BK013327	32/5	342, 3e−41
*Gymnadenia rhellicani*, GHXH01324014	31/7	331, 1e−33
*Ophrys sphegodes*, GHXJ01414654	30/5	341, 3e−33
*Ophrys fusca*, GHXI01129489	29/5	341, 1e−32
*Camellia reticulate*, GEER01003429	33/5	226, 6e−36
*Coriandrum sativum*, GGPN01001998	31/5	289, 5e−31
*Rhyncholacis* cf. penicillata Rhyc16, ICSC01000014	31/8	315, 8e−31
*Rhyncholacis* cf. penicillata Rhyc2783, ICSC01056734	32/5	372, 1e−30
*Silene dioica*, GFCG01071918	31/16	364, 1e−26
*Red clover RNA virus 1*, MG596242	31/5	311, 9e−34
*Arceuthobium sichuanense virus 3*, BK059270	30/10	231, 2e−29
*Striga hermonthica*, ICPL01009187	29/5	317, 9e−22

* Domain position means the domain starting amino acid residue in replicase.

## Data Availability

All data are available upon reasonable request.

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
