# Peer review of "Predicted Membrane-Associated Domains in Proteins Encoded by Novel Monopartite Plant RNA Viruses Related to Members of the Family Benyviridae"

_ijms, 2023, doi:10.3390/ijms241512161_

Round 1

Reviewer 1 Report

REVIEWER'S REPORT

Manucsript title:

 Predicted membrane-associated domains in proteins encoded by novel 2 monopartite plant RNA viruses related to members of the family Benyviridae (Authors:

 Sergey Y. Morozov, Alexander A. Lezzhov and Andrey G. Solovy).

In this manuscript, the authors describes the contituation research of their earlier works. In this study, they examined the gene architecture of Relovirid genomes in depth and predicted the structural characteristics of encoded non-replicative "orphan" hydrophobic polypeptides with movement roles.

This manusript, in my opinion, is properly done and written in fairly good English, and thus it may be published in this journal. Nonetheless, after reading the entire text, I get the feeling that the authors are writing in mechanical manner and more for themselves than for the readers, as the entire manuscript content is rather specialized and does not immediately engage the reader with the work's importance. If the article is presented in such manner, the circle of readers working in relevant subjects may be considerably reduced.

 To my mind, the Abstract should begin with a brief statement of the current work, such as, for instance, "As a continuation (or as a follow-up) of our recent studies, in this work we examined in greater detail...".

In introduction. The sentence (in lines 56-59) and the last sentence of this section should be paraphrased.

In Results. The first sentence (lines 71-72) should be paraphrased as " The NCBI non-redundant nucleotide sequence library currently contains the only four annotated Reclovirids. " In lines 77-78, the sentence should be written " We discovered a total of 27 virus species (Figure 1) infecting 78 flowering plants from 12 families of monocotyledonous and dicotyledonous plants..."

 At the start of section 2.1 (line 83), please add the OSF's full name along with its abbreviation in brackets. In lines 89-90, the sentence should be written " The gene organization of the 3'-terminal regions in reclovirid RNAs appears to be fairly variable in evolution."

In section 2.2, the sentence in lines 109-111 should be paraphrased as " The BLASTP search revealed that Reclovirid full-length replicases contain moderately comparable domains of 340-350 amino acids (Table 2, Figure 2), which yield negative CDD results."

 In Discussion. The sentence in lines 178-179, might be paraphrased " What characteristics of these proteins make them suitable candidates for viral RNR transmission from cell to cell?" In line 198, the sentence should be written as " Obviously, these hypotheses require more experimental validation." In line 202, "...may also contain..." should be replaced by "...may have...".  The last sentence (lines 215-217) should be paraphrased.

The conclusions must be given in a separate section, stressing the key points revealed in this work.    

The manuscript is written in reasonably decent English, while some of the manuscript text may be improved.    

Author Response

 To my mind, the Abstract should begin with a brief statement of the current work, such as, for instance, "As a continuation (or as a follow-up) of our recent studies, in this work we examined in greater detail...".

- It is done.

In introduction. The sentence (in lines 56-59) and the last sentence of this section should be paraphrased. 

- It is done.

In Results. The first sentence (lines 71-72) should be paraphrased as " The NCBI non-redundant nucleotide sequence library currently contains the only four annotated Reclovirids. " In lines 77-78, the sentence should be written " We discovered a total of 27 virus species (Figure 1) infecting 78 flowering plants from 12 families of monocotyledonous and dicotyledonous plants..."

  • It is corrected.

 At the start of section 2.1 (line 83), please add the OSF's full name along with its abbreviation in brackets. In lines 89-90, the sentence should be written " The gene organization of the 3'-terminal regions in reclovirid RNAs appears to be fairly variable in evolution."

  • It is done.

In section 2.2, the sentence in lines 109-111 should be paraphrased as " The BLASTP search revealed that Reclovirid full-length replicases contain moderately comparable domains of 340-350 amino acids (Table 2, Figure 2), which yield negative CDD results. "

  • It is done.

 In Discussion. The sentence in lines 178-179, might be paraphrased " What characteristics of these proteins make them suitable candidates for viral RNR transmission from cell to cell? " In line 198, the sentence should be written as " Obviously, these hypotheses require more experimental validation." In line 202, "...may also contain..." should be replaced by "...may have...".  The last sentence (lines 215-217) should be paraphrased.

  • It is corrected.

The conclusions must be given in a separate section, stressing the key points revealed in this work.    

  • It is done.

Reviewer 2 Report

The English is well written, but some long sentences should be shorted. 

Author Response

The written presentation and readability can be improved by correcting typological errors, well illustrating of some figures, simplifying language, and shortening complex sentence structures. 

- Corrections are included into the revised text.

  1. Fig. 1 reference (see Figure 5 from [2]) is necessary? Legend Line 45 showing change to shows.

- This reference is excluded.

  1. Fig.2 illustrate abbreviations in legend, such as VLRA, REP, and ORF.

- It is done.

  1. Line 58 Thus, it can be proposed that the single-component. Changing ‘can’ to ‘would’ seems better.
  • It is corrected.
  1. Shorten long sentences, for examples,

Line 59-62 long sentence: Indeed, analysis of the coding potential of these novel viral genomes has revealed one or more cistrons ranging in size from 40-80 to 60 about 600 codons, located in the 3’-terminal region of the genomic RNA, encoding proteins with predicted hydrophobic segments that are structurally diverse among Reclovirids and have no analogues in other plant RNA viruses.

  • It is done.

Line 173-177 long sentence: If the closely related plant Benyviridae-like viruses use movement gene blocks (BMB, TGB, or TCMB) that encode one, two or three small proteins with hydrophobic membrane-spanning segments as well as RNA helicase, the reclovirid cell-to-cell movement may be carried out by a wide variety of the hydrophobic “orphan” proteins (usually a single protein per virus).

  • It is done.

Line 32-37 long sentence. In addition to members of the genus Benyvirus, the former brunch includes several members of a family Benyviridae, particularly, Wheat stripe mosaic virus and Fern benyvirus, and a recently described group of “Tetra-cistron movement block (TCMB)-containing viruses” (Tecimovirids) coding for the TCMB movement gene module instead of triple gene block  (TGB) of movement genes found in members of the family Benyviridae. [2, 7] (Figure 1).

- It is done.

  1. Line 73. Carrot associated RNA virus 1 and Arceuthobium. ‘,’ before and.

- It is corrected.

  1. Line 79. Reference necessary? [2]

- The reference is excluded.

  1. Line 83. two ORFs (Open reading frame) in first appearance in paper. Delete (Open reading frame) in line 232.
  • It is done.